# Volatile Composition and Sensory Properties as Quality Attributes of Fresh and Dried Hemp Flowers (*Cannabis sativa* L.)

**DOI:** 10.3390/foods9081118

**Published:** 2020-08-13

**Authors:** Andrzej Kwaśnica, Natalia Pachura, Klaudia Masztalerz, Adam Figiel, Aleksandra Zimmer, Robert Kupczyński, Katarzyna Wujcikowska, Angel A. Carbonell-Barrachina, Antoni Szumny, Henryk Różański

**Affiliations:** 1Faculty of Biotechnology and Food Science, Wrocław University of Environmental and Life Sciences, Norwida 25, 50-375 Wrocław, Poland; antoni.szumny@upwr.edu.pl; 2Laboratorium Badań Toksykologicznych Lab4Tox Sp. z o.o., ul. Kruszwicka 24/66, 53-652 Wrocław, Poland; 3Institute of Agricultural Engineering, Wrocław University of Environmental and Life Sciences, Chełmońskiego 37-41, 51-630 Wrocław, Poland; klaudia.masztalerz@upwr.edu.pl (K.M.); adam.figiel@upwr.edu.pl (A.F.); zimmer.ola@gmail.com (A.Z.); 4Department of Environment, Animal Hygiene and Welfare, Wrocław University of Environmental and Life Sciences, Chełmońskiego 38C, 51-630 Wrocław, Poland; robert.kupczynski@upwr.edu.pl (R.K.); k.wujcikowska@gmail.com (K.W.); 5Departamento Tecnología Agroalimentaria, Universidad Miguel Hernández, Carretera de Beniel, 03312-Orihuela, Alicante, Spain; angel.carbonell@umh.es; 6Institute of Health and Economy, Carpathian State College in Krosno, Rynek 1, 38-400 Krosno, Poland; rozanski@rozanski.ch

**Keywords:** *Cannabis sativa* L., drying methods, essential oils, sensory evaluation

## Abstract

Flowers of hemp (*Cannabis sativa* L.) are widely used in cosmetics, food, and in the pharmaceutical industry. The drying process plays a key role in retention of aroma and also in the quality of products. Seven variants of hemp flower drying, including convection drying (CD), vacuum–microwave drying (VMD), and combined drying consisting of convective pre-drying followed by vacuum–microwave finishing drying (CPD-VMFD) were checked in this study. For each process, we applied the two-term model. Dried material was submitted to color and chromatographical assessments. Analyses of obtained essential oil showed the presence of 93 volatile compounds, predominantly β-myrcene, limonene, and β-(*E*)-caryophyllene, as well as α-humulene. Application of 240 W during VMD and 50 °C during CD gave the highest retention of aroma compounds, amounting to 85 and 76%, respectively, but with huge color changes. Additionally, sensory analysis proved that drying with a microwave power of 240 W provides a product most similar to fresh material.

## 1. Introduction

*Cannabis sativa* L. is an annual herbal plant of the cannabis species (*Cannabaceae*) that has been known and cultivated especially in Asia since ancient times [1]. Nowadays, it is already cultivated all over the world and has a wide range of applications, including food, dietary supplements, medicines, body care products, fuel, paper, and as a building material, as well as a role in textiles [2]. Hemp contains around 750 natural chemical compounds, which can be classified into different classes [3]. The abundance of chemicals in cannabis flowers stems from the biosynthesis, particularly of terpenes and cannabinoids in the extracellular secretory cavity, known as the trichome. The active substances are secreted into the trichomes to prevent damage to plant cells and are the first line of defense against the external environment [4].

The most common and basic technique for conserving herbs and retaining bioactive compounds is drying. The research shows that, depending on the choice of drying method and parameters, different chemical and biological activity of herbs is obtained, due to different content of chemical compounds in their composition [5]. The selection of the drying method has a major influence on the content of volatile essential oils present in herbs, as shown in earlier studies [6,7,8,9,10]. During the drying process, the following changes in the composition of the essential oil may occur—an increase or decrease in the concentration of volatile substances, or the formation of new chemical compounds [5]. The most commonly used drying method is convection drying (CD), a process using a continuous flow of hot air to remove moisture from the biological material [11]. With the development of technology, alternatives for better drying performance have emerged, such as vacuum–microwave drying (VMD) and combined methods consisting of convectional pre-drying followed by vacuum–microwave finishing drying (CPD-VMFD). The VMD method allows faster drying of the material while avoiding high temperatures [12]. Due to the high cost of production, an alternative drying method, CPD-VMFD, has been proposed due to the requirement of high vacuum during drying. This multi-stage drying process coupling the two CD and VMD methods together allows a satisfactory degree of drying of the material as it offers the advantages of the VMD method with the high performance that the CD method provides [11].

Recently, there has been a focus on the use of secondary metabolites from hemp flowers, which are characterized by low tetrahydrocannabinol (THC) content, which are known as nonpsychotropic cannabinoids, terpenoids, and flavonoids [13,14,15]. Essential oils extracted from hemp flowers are widely used in cosmetology as ingredients used in the production of creams, soaps, and shampoos, as well as in the food industry as aromas for alcoholic and non-alcoholic beverages and additives in baking [16]. Hemp essential oils have shown an interesting antimicrobial effect and can constitute an economic, effective antiseptic. They are therefore used to treat wounds and infections such as food poisoning and nosocomial infections, and can be used against antibiotic-resistant bacterial strains [17]. In addition, they are commonly used as insecticides [18,19], fungicides [20], and growth inhibitors for unwanted plants [21], and can be used in plant protection as a means of stopping plant diseases and pest attacks [19]. Cannabinoids contained in hemp also play a special role. They are responsible for the modulation of hunger/satiety and participate in peripheral metabolic reactions of the liver, fat, muscles, and anti-inflammatory reactions in blood cells [22]. Despite this, they can also cause side effects such as imbalance, hallucinations, nausea, and drowsiness [23].

Hemp can also be used as a source of food because of its health-promoting properties. Hemp seed oil is characterized by high levels of exogenous fatty acids (EFA) and polyunsaturated fatty acids (PUFA) [24]. The oil contains linoleic acid and L-linolenic acid, as omega-6 and omega-3 acids, which are optimal for nutrition because of their proportions (3:1, LA:LNA). Hemp seed oil is additionally enriched with gamma-linolenic acid (GLA), which makes the nutritional value ultimately higher than most seed oils. A properly balanced oil prevents excessive accumulation of certain metabolic products and also provides the necessary intermediaries for the body to work efficiently. The benefits of hemp seed oil as a food product and food supplement are that it can enrich the diet of the potential consumer [25].

The leading theme of the research was to define the composition of the volatile profile of hemp flowers and to verify how selected drying methods influence the profile of volatile compounds and the contribution of cannabinoids. For this purpose, we distilled the essential oil from the material under study and then analyzed it using gas chromatography in combination with the mass spectrometry technique (GC–MS).

## 2. Materials and Methods

### 2.1. Plant Material

Approximately 20 kg of hemp flowers var. Henola were harvested on 15 October 2019 from commercial field in Oborniki Śląskie (16°55′ E, 51°18′ N) Poland. The whole flowers, after being manually detached from the stem, were mixed and immediately subjected to drying, the distillation processes, as well as subsequent chemical analyses. The initial moisture content of the material was 68%_wb_ (wet basis), assessed by a vacuum dryer SPT-200 ZEAMIL (Horyzont, Krakow, Poland). Plant material was subjected to various drying processes, which were suspended when no changes in weight were observed. The voucher specimen of investigated flowers were deposited in local herbarium at the Department of Chemistry.

### 2.2. Drying Methods

In the study, three different drying methods were applied: convective drying (CD), vacuum–microwave drying (VMD), and combined drying consisting of convective pre-drying and vacuum–microwave finishing drying (CPD-VMFD). Approximately 60 g of hemp flowers were used in each case, and the process was carried out until the final moisture content of the sample was below 10%_wb_. This initial loading mass of sample allowed for thin layer drying without of the raw material compacting, which was necessary to meet the requirements for modeling the drying kinetics and to provide a sufficient amount of dry material for quality tests.

#### 2.2.1. Convective Drying (CD)

The CD was conducted on an apparatus located at the Institute of Agricultural Engineering (Wrocław University of Environmental and Life Sciences, Wrocław, Poland). The hemp flowers were placed in a special container (d = 100 mm) and dried in 50 °C, 60 °C, and 70 °C with an airflow of 0.5 ms^−1^. The preliminary tests revealed that that drying at temperatures below 50 °C takes too long to be applicable in industrial conditions in terms of energy consumption and thus operating costs, while temperatures above 70 °C lead to a drastic degradation of the chemical composition of the dried product. Due to difficulties in low temperature drying of hemp flowers, the temperatures used in this study were slightly elevated compared to the temperatures used for drying of some other herbal products such as thyme [6], sweet basil [8], or marjoram [7].

#### 2.2.2. Vacuum–Microwave Drying (VMD)

The VMD was performed using the SM 200 dryer (Plazmatronika, Wrocław, Poland). The samples were placed in a glass cylindrical drum that rotated at 6 rpm. The dryer was equipped with a BL 30P vacuum pump (Tepro, Koszalin, Poland), MP 211 vacuum manometer (Elvac, Bobolice, Poland), and a 0.15 m^3^ compensation tank. During the drying process, we used three power levels (240, 360, and 480 W) and pressure in the range of 4–6 kPa. Microwave powers were selected on the basis of the results from previous studies where similar materials were successfully dried in these conditions [6]. The maximum temperature of the dried hemp flowers was measured after removal from the dryer using an i50 infrared camera (Flir Systems AB, Stockholm, Sweden).

#### 2.2.3. Combined Drying Consisting of Convective Pre-Drying Followed by Vacuum–Microwave Finishing Drying (CPD-VMFD)

During combined drying, the samples were initially placed in drying baskets at the convective dryer for 60 min at 60 °C; then, the samples were moved to a vacuum–microwave dryer where finishing of drying at 360 W occurred until its final moisture content was below 10%_wb_. The time of convective pre-drying was proposed on the basis of preliminary studies, which showed that by that time water was effectively removed from the raw material at a satisfactory drying rate.

#### 2.2.4. Modelling of Drying Kinetics

Drying kinetics of hemp flowers were presented using a moisture ratio *MR* defined by relationship (1):(1)$$MR=\frac{M_{\left(t\right)}-M_{e}}{M_{0}-M_{e}}$$
where *M*_(*t*)_ is the moisture content of the sample at given time, *M*_0_ is the initial moisture content, and *M_e_* is an equilibrium moisture content that is usually omitted as the values of *M_e_* are relatively small (compared to *M*_(*t*)_ and *M*_0_), and therefore the simplified relationship (2) was used in the study without any significant influence on the drying kinetics modeling [26]:(2)$$MR=\frac{M_{\left(t\right)}}{M_{0}}$$

On the basis of obtained experimental data, we fitted several empirical drying models, including Newton, Midelli et al., logarithmic, two-term, and Page’s models, performed using TableCurve 2D software. The results of preliminary tests revealed that only the two-term equation (Equation (3)) can be considered, as it takes into account the best fit determined according to the highest values of *R*^2^ and the lowest values of root mean square error (RMSE).
(3)$$MR=a\cdot e^{-k_{1}\cdot t}+b\cdot e^{-k_{2}\cdot t}$$
where *k*_1_ and *k*_2_, and *a* and *b* denote drying constants and model coefficients, respectively.

### 2.3. Color Analysis

Color of the samples was measured in five repetitions using a Minolta Chroma Meter CR-400 (Minolta Co., Ltd., Osaka, Japan). The results were obtained in reference to International Commision on Illumination (CIE) *L***a***b** color space, where *L** stands for lightness, *a** values vary between negative (green) and positive (red), and *b** values vary between negative values indicated as blue and positive values indicated as yellow hues. The total change in color of dried material was expressed as DE* according to the following formula:$$\Delta E^{*}=\sqrt{{\left(L_{0}^{*}-L^{*}\right)}^{2}+{\left(a_{0}^{*}-a^{*}\right)}^{2}+{\left(b_{0}^{*}-b^{*}\right)}^{2}}$$
where *L*_0_*, *a*_0_*, and *b*_0_* denote the values of fresh material.

### 2.4. Distillation of Essential Oil (EO)

In the process of extraction of essential oils (EOs), we used the Deryng apparatus. A suitable quantity of weighed fresh or dried material was transferred to a 250 mL round-bottomed flask. The hemp flowers were poured with 100 mL of distilled water. The flask was placed in a heating mantel and the mixture was brought to boiling point and kept at this temperature for 45 min. When the boiling point was reached, we added 1 mL of cyclohexane to collect the essential oil, which contained 1 mg of 2-undecanone as internal standard (Sigma-Aldrich, Saint Louis, MO, USA). After the extraction process, organic phase with the essential oil was collected and stored at −18 °C until chromatographical analysis.

### 2.5. GC–MS Analyses

The profile of volatile compounds was analyzed using a gas chromatograph coupled with a mass spectrometer (Shimadzu GCMS QP 2020, Shimadzu, Kyoto, Japan). Separation was obtained by a capillary column Zebron ZB-5 (30 m, 0.25 mm, 0.25 μm; Phenomenex, Torrance, CA, USA). The GC–MS analysis was carried out according to the following parameters: scanning in the range from 35 to 320 m/z in electron ionization mode at 70 eV, in the option of 3 scans s^−1^. Analyses were performed using helium as a carrier gas at a flow rate of 1.11 mL min^−1^ in a split ratio of 1:20. The GC oven temperature was programmed from 45 °C as initial temperature to 150 °C at a rate of 2 °C/min, then to 270 °C at a rate of 15 °C and kept for 5 min.

Identification of compounds were based on 3 independend methods: (a) comparison of obtained spectra with databases NIST 17 (National Institute of Standards and Technology) [27] and FFNSC (Mass Spectra of Flavors and Fragrances of Natural and Synthetic Compounds) [28], (b) comparison of calculated retention indices (RI) using a retention indices calculator [29] with values presented in NIST 17 and FFNSC, and (c) comparison of retention times of unknown compounds with authentic standards. For comparison of mass spectra, we used the AMDIS (v. 2.73) and GCMS solution (v. 4.20) programs. Additionally, all experimental RI were compared with those published in Adams [30].

### 2.6. Sensory Evaluation

The intensity of the main sensory attributes of dried hemp flowers were evaluated by a trained sensory panel. The panel consisted of 7 panelists (4 males and 3 females), aged between 34 and 53 years old. Panelists belonged to the research group “Food Quality and Safety” of the Universidad Miguel Hernández de Elche (UMH) and had over 1000 h of evaluation experience. The panel was selected and trained following the International Organization for Standardization ISO standard 8586-1 (1993), and it is specialized in descriptive sensory evaluation of fruits and vegetables and has a wide expertise in studying the effects of drying on different matrixes, such herbs, fruits, vegetables, and mushrooms [31].

Descriptive sensory analysis (DSA) was used to describe the dried hemp flowers. During 1 orientation session of 90 min, panelists discussed about the main odor (perception of volatile compounds with the product outside the mouth) and agreed on their use of key attributes/descriptors. Panelists agreed that the sensory profile of these dried samples could be described using 11 attributes: (i) fresh material: hemp flower ID, fresh vegetables, citrus, balsamic, spicy, and anise odor; and (ii) dried material: cooked, hay-woody, camomile, earthy, and burnt. Reference products of these attributes, with intensity similar to those of the samples under evaluation were prepared and provided to the panel.

The evaluation was performed in normalized individual booths with controlled illumination and temperature, 23 ± 2 °C. Samples coded with 3-digit random numbers were randomly presented to each panelist in odor-free plastic beakers of 100 mL with lids; samples were left for 15 min at room temperature prior to analysis. The 9 samples under analyses were analyzed in 3 sessions in which 3 samples per session were randomly presented to the panel; this design was selected to avoid sensory fatigue. The intensity of the sensory attributes was scored using a scale from 0 to 10, where 0 = none or not perceptible intensity and 10 = extremely high intensity.

### 2.7. Statistical Analysis

All analyses were performed using the STATISTICA 13.3 software (StatSoft, Krakow, Poland). Data are expressed as means values ± standard deviation. Before analyses, all data were screened for normality using the Shapiro–Wilk test. The results from drying kinetics were subjected to the analysis of variance using Tukey’s test (*p* < 0.05) and the results from analyses of essential oils were subjected to the analysis of variance using Duncan’s test (*p* < 0.05). The data from the sensory analysis were also subjected to the Honest Significance Difference (HSD) Tukey’s test (*p* < 0.05).

## 3. Results and Discussion

### 3.1. Drying Methods

Figure 1 shows the drying kinetics of hemp flowers treated by convective drying (CD) (a), vacuum–microwave drying (VMD) (b), and combined drying (CPD-VMFD) (c), whereas Table 1 presents model constants with drying times and maximum temperature during drying using different methods.

The two-term model was used to describe the drying kinetics of hemp flowers. The good fit of this model was confirmed by high values of coefficient of determination (*R*^2^ > 0.98 for all the drying methods) and low values of RMSE (Table 1). The model was previously successfully used in studies on pomegranate rind and arils [32] and ginger slices [33]. This was due to the model structure that specifically describes two stages of drying, with coefficient *a* referring to the breadth of the first stage of drying where intensive evaporation occurs, illustrated by high values of drying constant *k*_1_, and the second stage (coefficient *b*), with decreasing drying rate characterized by low values of drying constant *k*_2_. On the other hand, 1/*k*_1_ and 1/*k*_2_ are time constants that express the time required for 37% decrease of *a* and *b* values, respectively [34]. Therefore, it can be stated that the higher the value of the drying constant *k*_1_ or *k*_2_, the lower the value of the time constant 1/*k*_1_ or 1/*k*_2_ and thus the shorter the time of the relevant stage of drying.

It is worth mentioning that during the first stage of drying, the mass transfer mainly consists of water evaporation from the surface of the material at a high drying rate assured by sufficient water diffusion at a relatively high external moisture content, whereas during the second stage of drying, the mass transfer occurs at a lower drying rate limited by internal water diffusion hindered by decreasing moisture content [5]. The drying kinetics of CD might be affected to some extend by shrinkage of the dried material manifested by curling of the whole flower at the end of the first stage of drying, leading to a decrease of surface evaporation and thus hindering the mass transfer, which results in the reduction of drying rate. It can be seen that the change of temperature of hot air during convective drying (from 50 to 70 °C) results in an increase of drying rate followed by higher values of *k*_1_ and *k*_2_ constants (increase from *k*_1_ = 0.02469 min^−1^ in case of CD50 to *k*_1_ = 0.06093 min^−1^ in the case of CD70 during first stage of drying). Similar behavior was previously reported in studies on orange slices [35] and red pepper [36].

As for VMD, it should be noted that values of coefficient *a* were very high (over 0.9), which means that most of the drying occurred in the first stage, while coefficient *b* was over nine times lower than coefficient *a*, which marks shorter period of the second stage of drying. Moreover, with higher microwave power, *k*_1_ constant increased, which shows that the drying rate was greatly influenced by the power of magnetrons in the first stage of drying. However, in the second stage of drying, the drying rate decreased with an increase of microwave power. It was due to the extended first phase of the process (defined by coefficient *a*) with high *k*_1_ values that mostly contributed to the water evaporation and reduction of moisture content and thus moisture ratio (MR). Still, the second stage greatly affected the drying kinetics and the time of the process, even though it had little impact on MR values, which decreased at low rate. The course of drying curves showed that application of 480 W instead of 360 W did not contribute to an increase of the drying rate at the initial phase of drying, which was confirmed by similar values of *k*_1_ amounting to 0.1157 and 0.1233, respectively. This means that from a practical point of view, starting with a microwave power higher than 360 W is not reasonable during VMD of hemp flowers. However, the elevation of microwave power from 360 to 480 W is reasonable after the initial phase of VMD lasting around 8 min (Figure 1b). It can be presumed that during the initial phase of VMD at high microwave power, the water removal rate was restricted by the increased resistance of mass transfer caused by the thickening of the water molecules in the outer layer of the material as a result of a transport mechanism of the Darcy type [37] that had too much intense pressure at the limited possibility of water evaporation from the surface of the dried material. From this perspective, shrinkage of the dried material described above additionally hindered water evaporation from the decreased surface area and promoted the thickening of the water molecules in the outer layer of the material.

In case of CPD-VMFD, convective pre-drying can be easily described by the model parameters fitted and adjusted for CD at 60 °C; thus, only vacuum–microwave finishing drying is provided in Table 1. During combined drying, the most intensive evaporation occurred throughout the pre-drying period, and therefore the joint contribution of finishing drying (defined as the sum of *a* and *b* coefficients) was quite low. However, the drying constant *k*_1_ reached the highest value (0.12680 min^−1^) among all applied drying methods, which was almost 10% more than in the case of the sole use of VMD360, while *k*_2_ was also very high and was only lower than in the case of VMD360. The relatively high values of *k*_1_ and *k*_2_ may be somewhat surprising, taking into account the fact that VMFD started when the surface of plant material had been already reduced due to the shrinkage formed during CPD. This can be explained by the recovery of the water molecule distribution within the entire volume of the pre-dried material during the time necessary to reload the sample.

Obtained data show that an increase of drying temperature resulted in shorter drying times during CD. An increase by 20 °C (from 50 to 70 °C) resulted in 39% time reduction (from 840 to 510 min), which is consistent with previous studies on true lavender leaves [38], carrot slices [39], and cornelian cherry fruit [40]. Moreover, VMD was much shorter than CD. This was due to the volumetric heating occurring during VMD that sped up the process by increasing the temperature inside of the sample [41,42]. As a result, samples treated by VMD at 480 W were dried 21 times faster when compared to CD at 50 °C. Furthermore, the power of magnetrons during VMD influenced water removal and resulted in shorter processing times when higher powers were applied [43]; namely, an increase from 240 W to 480 W decreased the time of drying by 72 min (reduction from 112 to 40 min for VMD at 240 W and VMD at 480 W, respectively).

On the other hand, combined methods (CPD-VMFD) resulted in a much shorter time of drying compared to CD, yet were still longer than VMD, which is consistent with previous studies on pomegranate arils [32] and quinces [44]. This method proved to be beneficial in other studies, i.e., on sweet basil [8] and thyme [6], where high quality products were obtained in a significantly shorter time period (compared to CD).

It is worth noting that the duration of VMD did not depend on the maximal temperature (T_Max_) of the samples treated by microwaves. Usually the maximal temperature is achieved by VMD samples at the end of drying when the heat energy generated by water dipoles in the microwave field are in excess the energy necessary for water evaporation [45]. In the case of VMD of hemp flowers, the highest values of T_Max_, amounting to 61 and 59 °C, were obtained at 480 and 240 W, respectively. The relatively high value of T_Max_ for samples dried at 240 W can be explained by the longer exposition to microwave radiation and thus the accumulation of the heat energy delivered at low microwave power [9]. The lowest values of T_Max_, amounting to 61 and 59 °C, were found for VMD360 and CPD-VMFD, respectively. During both the drying protocols, the same microwave power of 360 W was applied, which suggests that the mean value of microwave wattage maintained the thermal energy balance at the lowest temperature of VMD material. Generally, the temperature of VMD herbal products is relatively low due to their specific morphological structure, preventing the excessive increase of inner pressure caused by microwave heating [6].

It can be seen that all the applied drying protocols differed in terms of the form of heat energy delivery, duration, and dried material temperature. Therefore, it is crucial to evaluate these drying protocols also in terms of their impact on the color, chemical composition, and sensory attributes of hemp flowers.

### 3.2. Color Analysis

Color of the samples is presented in the Table 2. The drying process increased lightness, yellowness, and greenness of the samples, which was confirmed by higher values of parameters *L** and *b** and lower values of *a** parameter. The lightest samples (highest *L** values) with yellowest hues (one of the lowest values of *a** parameter) were obtained after being subjected to VMD at 480 W. On the other hand, samples of VMD at 240 W were significantly darker (lowest *L**) than the samples obtained by other drying methods. Moreover, VMD at 360 W resulted in the greenest samples (highest values of *a**) and lowest values of *b** parameter, which had yellow hues. Generally, an increase of temperature during CD resulted in the obtaining of darker samples with a higher share of green and yellow hues, despite a slight decrease in *a** and *b** values. Although a similar effect on greenness and yellowness was observed when increasing the microwave power during VMD, these color alterations were associated with lightening of the samples. It is worth noting that the values of the color parameters determined for CPD-VMFD samples were between the relevant values obtained for samples dried by CD at 60 °C and VMD at 360 W. The significance of the color changes can be estimated by the values of total color change Δ*E**. According to Sumic at al. [46], if Δ*E** is less than 1.0, it is assumed that the difference in color would not be perceptible by the human eye. The total color change of the dried samples was much higher than 1.0 for all cases. However, the lowest change of 19.2 was stated for the VMD240 sample, whereas the highest change of 24.20 was found for the VMD480 sample. This indicates that microwave power during VMD had a higher impact on the color change of the dried material than hot air temperature during CD.

### 3.3. Volatile Constituents of Fresh Cannabis sativa Flowers

Analysis of GC–MS of the distilled essential oils from the hemp flower revealed 93 peaks, which were considered to be volatile substances (chromatograms from the GC–MS analysis of volatile compounds content of hemp flowers are available in Appendix A). These compounds are collected in Table 3. Of all the volatile components identified in the tested samples of hemp flowers, we could distinguish the following main substances: β-myrcene (26.66%), β-(*E*)-caryophyllene (17.50%), limonene (10.45%), α-humulene (7.26%), caryophyllene oxide (3.79%), β-pinene (2.51%), terpinolene (2.50%), and α-pinene (2.16%). In smaller quantities, (E,E)-α-farnesene (1.80%), α-selinene (1.65%), β-chamigrene (1.41%), humulene epoxide II (1.25%), pseudowiddrene (1.24%), and β-trans-ocimene (1.12%) were identified. The substances above have a considerable influence on the fragrance quality of the hemp flowers.

In previous studies, similar results were found, where the identified compounds were as follows: β-(*E*)-caryophyllene (23.8%), α-pinene (16.4%), and β-myrcene (14.2%). The remaining compounds worth showing had the following percentages: terpinolene (9.6%), α-humulene (8.3%), β-pinene (5.2%), β-(*E*)-ocimene (5.1%), and β-(*E*)-farnesene (3.0%) [47]. Benelli in other studies also showed a similar profile of volatile components to those mentioned above, where β-(E)-caryophyllene (21.4%), β-myrcene (11.3%), α-pinene (7.8%), terpinolene (7.6%), α-humulene (7.1%), β-(*E*)-ocimene (3.9%), and β-pinene (2.9%) were identified as the most representative compounds [48].

Considering the analysis of the obtained fractions, we observed a similarity with the paper published by Nissen. Depending on the cannabis cultivar, the following substances can be distinguished, β-myrcene (12.46–29.22%), α-pinene (10.9–16.99%), β-(*E*)-caryophyllene (10.56–13.90%), β-pinene (6.38–9.33%), α-humulene (4.84–6.71%), terpinolene (3.42–10.73%), limonene (3.11–4.99%), and β-(*E*)-ocimene (2.03–9.34%) as those occurring in the highest amount [17]. Mediavilla and Steinemann also studied the profile of volatile compounds on different cannabis strains. The main compounds isolated were β-myrcene (29.4–65.8%), β-(*E*)-caryophyllene (3.8–37.5%), α-pinene (2.3–31.0%), terpinolene (0.4–23.8%), β-(*E*)-ocimene (0.3–10.2%), α-humulene (0.7–7.9%), β-pinene (0.6–7.8%), and limonene (0.2–6.9%) [49]. Iseppi et al. reported β-myrcene (4.5–39.2%), β-(*E*)-caryophyllene (8.5–29.8%), β-pinene (4.8–25.4%), terpinolene (1.9–9.6%), α-pinene (3.4–8.2%), β-(*E*)-ocimene (2.2–7.1%), α-humulene (2.2–6.6%), and limonene (0.1–5.7%) as the main volatile components of hemp flower essential oils. These discrepancies may result from the use of different cultivars for research and from the condition of raw material [50].

### 3.4. Effects of Different Drying Methods on the Volatile Compound Content in Cannabis sativa

In the conducted studies of the biological material consisting of fresh hemp flowers, we found that the content of essential oil was 0.21 g/100 g^−1^ of dry weight (DW). Considering the yield of essential oils in cannabis flowers, we can conclude that it was high, as 0.1–0.25% [17] and 0.25% [51] of essential oils were reported in previous studies. Furthermore, Liang et al. [52] and Chalchat and Özcan [53] reported the yield of essential oils in the inflorescences of sage and basil as 0.2% and 0.5%, respectively. A comparison of changes in the content of 11 main volatile components in hemp with reference to different drying methods is shown in Table 4. As far as the total content of essential oils is concerned, it is not analogous to the 11 main compounds, as they represent 77.88% of the total.

By analyzing the content of essential oils in relation to different drying processes, we observed that each of them had an impact on its final yield. VMD at 240 W (0.18 g/100 g^−1^) turned out to be the most effective drying method, followed by CD at 50 °C (0.16 g/100 g^−1^), while VMD at 360 W and VMD at 480 were the next methods with the same oil yield (0.15 g/100 g^−1^). The least effective method was CPD-VMFD, with an essential oil yield of only 0.11 g/100 g. The above data are also presented as percentage oil recovery, which is included in Table 3. Other works describing the drying of marjoram [7] and thyme [6] also prove that the VMD at 240 W is the best solution to stop volatile compounds. Vacuum–microwave drying has also proven to be the most effective method for oregano drying [9]. However, this is in contradiction with the paper by Łyczko’s et al. [54] that described the drying of lavender flowers, where this method was proven to be the least effective. However, the separate systematics of hemp and lavender flowers must be taken into consideration.

In addition, the study also showed the relationship between the intensity of the treatment and the losses that occur. It was observed that with increasing intensity of drying conditions, the level of losses of volatile compounds in CD increased in comparison to VMD. In CD, the loss of volatile compounds was recorded from 0.16 to 0.12 (decrease by 19.01%), with an increase in drying temperature from 50 °C to 70 °C, and with an increase in power from 240 W to 480 W in VMD, the concentration decreased from 0.18 to 0.15 (decrease by 14.29%). Sanchez et al. [8] in research on sweet basil also reported this dependence. As far as the CPD-VMFD method is concerned, the losses of volatile compounds compared to other drying methods were highest, as a 48% decrease in the volatile compounds in essential oil was observed. Consequently, this method is not a recommended drying method because it does not improve the aroma quality of dried hemp flowers. A similar situation was documented during research into drying shiitake mushrooms [55], where the combined drying method proved to be the least effective. This shows that VMD is a better method for drying cannabis flowers if the aim is to preserve as much essential oil as possible.

It is also worth noting the individual compounds of the 11 main ingredients. For example, to preserve the main fragrance compounds in the cannabis flower such as β-myrcene and β-(E)-caryophyllene, CD at 70 °C (for β-myrcene) and CD at 50 °C (for β-(E)-caryophyllene) proved to be the best drying methods.

It is also important to note the increase of α-pinene from 2.16% to 11.13% (CPD-VMFD) and caryophyllene oxide from 3.79% to 12.16% (VMD at 480 W), as well as the decrease of β-myrcene from 26.66% to 7.54% (VMD at 240 W). The direction of percentage changes of individual compounds resulted from their different susceptibility to the thermal degradation as an effect to an irreversible oxidation process and volatilization enhanced by water evaporation, leading to a decrease of the dry matter content. This shows that all drying protocols had a considerable impact on the final ratio of the mass of individual components with different retention ability to the reduced mass of the dry matter in the hemp flower.

### 3.5. Sensory Value of Cannabis sativa Flowers According to Various Drying Methods

Only odor descriptors (perception of volatile compounds with the product outside the mouth) were evaluated, and the protocol followed was similar to that described previously to other dried herbs [7] and mushrooms [31]. Six descriptors related to fresh plant material (hemp ID: aromatics associated with fresh hemp flowers, and fresh vegetables: aromatics associated with fresh but non-identified vegetables) were assayed together with five that were mostly related to the drying process (e.g., hay-woody and burnt). The nine samples of hemp dried flowers were analyzed in three sessions in which three samples per session were randomly presented to the panel. The results presented in Table 5 clearly showed that the sample that kept most of the intensity of the fresh material was that obtained using VMD at 240 W, followed by that prepared using the combined method at 50 °C.

In general, with the increasing strength of the drying process, i.e., the temperature in the CD and the power in the VMD, the sensory quality of the dried samples deteriorates due to significant losses of key odor compounds related to the fresh plant material. The sensory results agreed with previously discussed trends for the total content of volatile compounds.

## 4. Conclusions

The results of the study revealed that the drying kinetics of hemp flowers treated by convective drying (CD), vacuum–microwave drying (VMD), and combined drying composed of convective pre-drying followed by vacuum–microwave finish drying (CPD-VMFD) can be satisfactory described using a two-term empirical model. During the drying process, we found loses in 93 analyzed volatiles from 48% for CPD-VMFD to 15% for VMD at 240 W, which was finally chosen as optimal for retention of aroma-active compounds. In that variant, a significant decrease of β-myrcene was observed. Taking drying time into consideration, the shortest dehydration operation was VMD at 480 W (40 min) in contrast to CD at 50 °C (840 min), although the loses of compounds were around 30%. From a sensory point of view, the best drying treatment was VMD at 240 W, because it produced dried samples most resembling the fresh material, with high intensities of key sensory descriptors such as hemp flower ID, fresh vegetables, citrus, balsamic, and anise. Unfortunately, that process produced the most changes in flower color. Although combined drying (CPD-VMFD) could be advantageous from a practical point of view, too much degradation of the chemical composition identified in the raw material prevents this method to be applied to a greater extent. Taking into account the influence of individual drying conditions on the drying time and quality parameters of the dried product, VMD at 240 W can be recommended by the industry as the best option for hemp flower drying.

## Figures and Tables

**Figure 1 foods-09-01118-f001:**
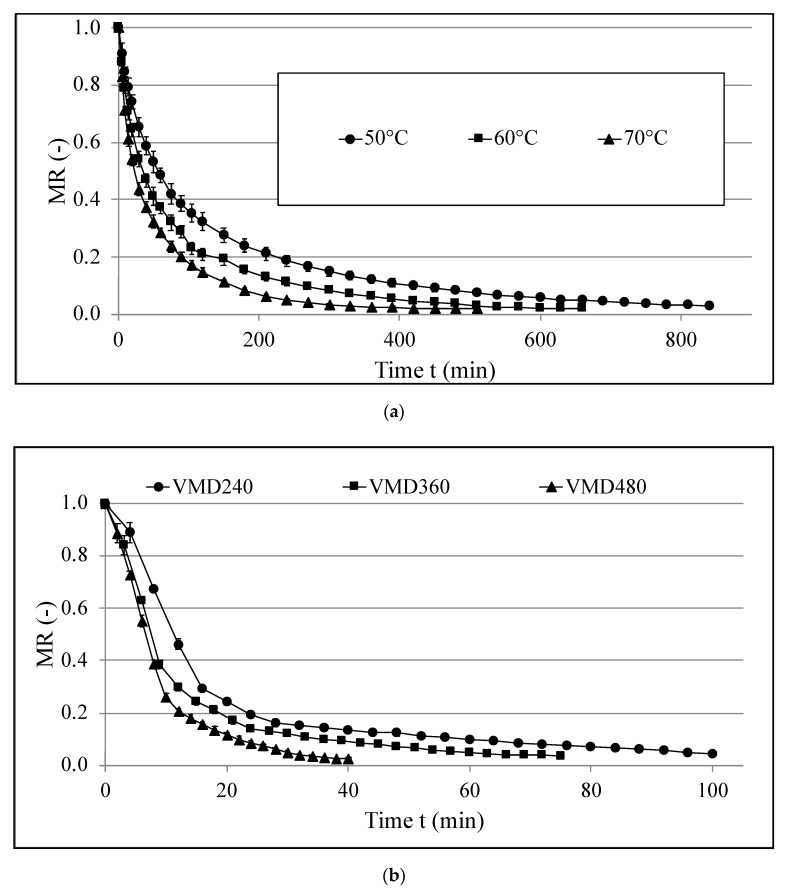
Drying kinetics of hemp flowers treated by convective drying (CD) at 50, 60, and 70 °C (**a**); vacuum–microwave drying (VMD) at 240, 360, and 480 W (**b**); and combined drying consisting of convective pre-drying (CPD) at 60 °C for 1 h and following vacuum–microwave finishing drying (VMFD) at 360 W (**c**).

**Table 1 foods-09-01118-t001:** Model parameters (*a*, *k*_1_, *b*, *k*_2_); maximum temperature of the hemp flowers during drying (TMax); and drying times t_CD_ and t_VMD_ of CD and VMD, respectively.

Drying Methods	Parameters				Statistics		Drying Time (min)		T_Max_ (°C)
	*a*	*k*_1_, (min^−1^)	*b*	*k*_2_, (min^−1^)	RMSE	*R* ^2^	t_CD_	t_VMD_	
CD50	0.54412	0.02469	0.44193	0.00350	0.00537	0.9997	840	-	50
CD60	0.58617	0.03623	0.40373	0.00521	0.00692	0.9995	660	-	60
CD70	0.53544	0.06093	0.46011	0.00912	0.00810	0.9993	510	-	70
VMD240	0.95054	0.08296	0.11951	0.00742	0.03445	0.9827	-	112	59
VMD360	0.91787	0.11572	0.12779	0.01484	0.02596	0.9903	-	78	54
VMD480	1.04449	0.12333	0.01818	0.00014	0.03465	0.9883	-	40	61
**CPD-VMFD**	**0.29965**	**0.12680**	**0.06519**	**0.01216**	**0.00785**	**0.9938**	**60**	**54**	**55**

**Table 2 foods-09-01118-t002:** Color parameters of hemp flowers subjected to drying using different methods.

Drying Method	Color			
	*L**	*a**	*b**	Δ*E**
Fresh	20.61 ± 5.37	−1.98 ± 0.61	8.41 ± 2.87	-
CD50	42.48 ± 1.59 ^a,b,1^	−4.49 ± 0.83 ^a,b^	12.06 ± 1.01 ^a,b^	22.31
CD60	41.88 ± 0.7 ^a,c^	−4.99 ± 0.6 ^a^	10.77 ± 0.4 ^a,c^	21.61
CD70	41.72 ± 1.26 ^a,c^	−4.82 ± 0.25 ^a^	11.25 ± 0.72 ^a,b^	21.49
VMD240	39.78 ± 0.43 ^c^	−3.57 ± 0.66 ^b^	9.6 ± 0.34 ^c^	19.27
VMD360	43.56 ± 1.28 ^a,b^	−3.5 ± 0.62 ^b^	11.34 ± 0.61 ^a,b^	23.19
VMD480	44.27 ± 0.93 ^b^	−4.83 ± 0.37 ^a^	12.62 ± 0.57 ^b^	24.20
CPD-VMFD	42.09 ± 1.09 ^a,b^	−4.59 ± 0.52 ^a,b^	10.99 ± 1.14 ^a,c^	21.79

^1^ Mean values followed by the same letter were not significantly different (*p* < 0.05) according to the HSD Tukey’s least significance difference test.

**Table 3 foods-09-01118-t003:** Complete volatile constituents of fresh *Cannabis sativa* flowers.

Compound	RT (min)	Retention Indices (RI)				Content (%) ^5^
		RI_lit ^1^	RI_lit ^2^	RI_lit ^3^	RI_exp ^4^	
**Octane**	4.585	800	800	800	800	0.08 ± 0.01
**(2*E*)-2-Hexenal**	6.130	855	851	850	855	tr ^6^
**(3*Z*)-3-Hexen-1-ol**	6.240	859	857	853	857	0.06 ± 0.02
**1-Hexanol**	6.705	870	868	867	869	tr
**2-Heptanone**	7.485	892	891	892	890	0.05 ± 0.01
**Heptanal**	7.900	902	901	906	900	0.51 ± 0.12
**(2*E*,4*E*)-2,4-Hexadienal**	8.395	909	911	914	909	tr
**Artemisia triene**	8.620	929	929	922	926	0.50 ± 0.21
**α-Thujene**	9.035	930	929	927	928	0.06 ± 0.08
**α-Pinene**	9.310	939	937	933	932	2.16 ± 0.82
**Fenchene**	9.985	952	952	948	953	0.25 ± 0.05
**Benzaldehyde**	10.585	960	962	960	960	tr
**(2*E*)-2-Hepten-1-ol**	11.025	965	978	964	970	tr
**Sabinene**	11.290	975	974	972	976	tr
**β-Pinene**	11.400	979	979	978	976	2.51 ± 0.32
***trans*-Isolimonene**	11.815	984	983	984	983	tr
**6-Methyl-5-hepten-2-one**	12.115	985	985	986	987	tr
**β-Myrcene**	12.455	990	991	991	994	26.66 ± 1.79
**α-Phellandrene**	12.970	1002	1005	1007	1004	0.13 ± 0.06
**3-Carene**	13.280	1011	1011	1009	1007	0.12 ± 0.09
**α-Terpinene**	13.665	1017	1017	1018	1017	0.12 ± 0.03
**o-Cymene**	13.910	1026	1022	1024	1021	tr
**p-Cymene**	14.115	1024	1030	1025	1025	tr
**Limonene**	14.415	1029	1030	1030	1030	10.45 ± 1.21
**Sylvestrene**	14.745	1030	1027	1031	1035	tr
**β-*cis*-Ocimene**	15.055	1037	1038	1035	1041	0.09 ± 0.03
**β-*trans*-Ocimene**	15.660	1050	1049	1046	1051	1.12 ± 0.14
**Prenyl isobutyrate**	15.930	1052	1052	1050	1052	tr
**Oct-(3*Z*)-enol**	16.025	1054	1059	1059	1054	tr
**γ-Terpinene**	16.205	1059	1060	1058	1062	0.14 ± 0.07
***cis*-Sabinene hydrate**	16.695	1070	1070	1069	1071	0.14 ± 0.09
**2-*trans*-Octenol**	17.285	1074	1072	1073	1075	0.05 ± 0.01
**Terpinolene**	18.030	1088	1088	1086	1087	2.50 ± 0.43
**6,7-Epoxymyrcene**	18.445	1092	1090	1096	1091	0.06 ± 0.02
***trans*-Sabinene hydrate**	18.650	1094	1093	1099	1096	tr
**Linalool**	18.895	1096	1099	1101	1099	0.09 ± 0.11
**Nonanal**	19.150	1100	1104	1107	1105	0.32 ± 0.14
**Fenchol**	19.565	1116	1113	1119	1116	0.82 ± 0.03
***trans*-Pinene hydrate**	20.025	1122	1120	1121	1118	0.82 ± 0.21
***cis*-Pinene hydrate**	21.340	1143	1143	1144	1139	0.11 ± 0.07
**β-Terpineol**	21.720	1144	1144	1149	1145	0.06 ± 0.01
**Ipsdienol**	21.875	1145	1147	1146	1147	0.1 ± 0.02
**Myrcenone**	22.000	1149	1145	1149	1150	0.09 ± 0.10
**α-Pinene oxide**	22.635	1159	1157	1156	1158	0.09 ± 0.05
**3-Thujanol**	23.005	1168	1167	1169	1165	0.09 ± 0.03
**Terpinen-4-ol**	23.740	1177	1177	1177	1177	0.12 ± 0.09
**Isogeranial**	23.880	1180	1182	1179	1177	tr
**α-Terpineol**	24.670	1188	1189	1195	1189	0.43 ± 0.12
**Hexyl butanoate**	25.035	1192	1192	1195	1194	0.14 ± 0.07
***trans*-4-Caranone**	25.530	1196	1197	1200	1196	tr
**Bornyl acetate**	30.975	1285	1285	1285	1282	tr
**α-Cubebene**	35.095	1351	1351	1349	1345	tr
**α-Ylangene**	36.410	1375	1372	1371	1368	0.12 ± 0.01
**α-Copaene**	36.700	1376	1376	1375	1373	tr
**Hexyl hexanoate**	37.405	1383	1384	1387	1383	tr
**7-*epi*-Sesquithujene**	37.705	1391	1402	1389	1387	0.07 ± 0.03
**Isocaryophyllene**	38.610	1408	1406	1405	1402	0.54 ± 0.12
**α-Gurjunene**	38.800	1409	1409	1406	1404	tr
**β-(*E*)-Caryophyllene**	39.480	1419	1419	1424	1415	17.50 ± 1.75
**β-Duprezianene**	39.735	1422	1422	1427	1420	0.09 ± 0.03
**α-*trans*-Bergamotene**	40.560	1434	1435	1432	1435	0.17 ± 0.09
**β-Humulene**	40.660	1438	1440	1440	1435	0.11 ± 0.05
**Guaia-6,9-diene**	40.925	1444	1443	1444	1440	0.17 ± 0.11
**α-Humulene**	41.485	1454	1454	1454	1453	7.26 ± 1.48
**Khusimene**	41.650	1455	1451	1451	1455	0.21 ± 0.14
**β-(*E*)-Farnesene**	41.870	1456	1457	1452	1457	0.33 ± 0.21
**9-*epi*-(*E*)-Caryophyllene**	41.985	1464	1466	1464	1459	0.39 ± 0.18
**Dodec-(2*E*)-enal**	42.380	1466	1468	1469	1464	0.15 ± 0.08
**γ-Gurjunene**	42.865	1474	1475	1476	1473	0.10 ± 0.01
**β-Chamigrene**	43.290	1477	1476	1479	1479	1.41 ± 0.56
**γ-Selinene**	43.425	1479	1479	1480	1481	0.62 ± 0.19
**α-Selinene**	43.990	1496	1494	1495	1490	1.65 ± 0.77
**α-Zingiberene**	44.330	1498	1495	1496	1495	0.31 ± 0.16
**δ-Amorphene**	44.785	1504	1505	1506	1499	0.38 ± 0.09
**(*E,E*)-α-Farnesene**	45.100	1505	1508	1504	1509	1.80 ± 0.64
**Pseudowiddrene**	45.385	1509	1510	1510	1512	1.24 ± 0.57
**δ-Cadinene**	45.755	1523	1524	1518	1514	0.21 ± 0.03
**γ-Cuprenene**	46.320	1533	1532	1530	1532	0.56 ± 0.12
**Selina-4(15),7(11)-diene**	46.575	1546	1542	1540	1533	2.13 ± 0.86
**α-Cadinene**	46.705	1538	1538	1538	1538	0.82 ± 0.31
**(*E*)-α-Bisabolene**	47.025	-	1512	1540	1538	0.28 ± 0.16
***cis*-Muurol-5-en-4-β-ol**	47.300	1551	1549	1548	1548	0.07 ± 0.03
**Germacrene B**	47.555	1561	1557	1557	1551	0.16 ± 0.05
**Lippifoli-1(6)-en-5-one**	47.995	1552	1553	1551	1557	0.69 ± 0.32
***epi*-Longipinanol**	48.310	1563	1566	1558	1564	0.44 ± 0.11
**Caryophyllene oxide**	49.055	1583	1581	1581	1576	3.79 ± 0.44
**Humulene epoxide I**	49.965	-	1604	1604	1590	0.18 ± 0.05
**Humulene epoxide II**	50.540	1608	1606	1613	1607	1.25 ± 0.14
**1,10-di-*epi*-Cubenol**	51.115	1619	1615	1614	1612	0.62 ± 0.23
***epi*-γ-Eudesmol**	51.455	1623	1622	1624	1623	0.64 ± 0.45
**α-Acorenol**	51.935	1633	1631	1632	1633	0.14 ± 0.04
**Caryophylla-4(12),8(13)-dien-5α-ol**	52.350	1640	1637	1642	1646	0.80 ± 0.23
**α-Bisabolol**	54.140	1685	1684	1688	1686	0.44 ± 0.13

Retention indices according to ^1^ Adams [30], ^2^ NIST 17 database [27], ^3^ FFNSC [28]; ^4^ % calculated from TIC data; 5 experimental retention indices calculated against n-alkanes; 6 tr < 0.05%.

**Table 4 foods-09-01118-t004:** Comparison of changes in volatile composition of fresh and dried hemp flowers.

Compound	Drying Method							
	Fresh ^1^	CD 50 °C	CD 60 °C	CD 70 °C	VMD 240 W	VMD 360 W	VMD 480 W	CPD-VMFD
	Content (%)							
**α-Pinene**	2.16 ^a,3^	10.79 ^g^	7.16 ^e^	8.27 ^e^	3.67 ^b^	5.86 ^d^	9.58 ^f^	11.13 ^g^
**β-Pinene**	2.51 ^a^	4.47 ^a^	3.49 ^c^	4.43 ^d^	2.17 ^a^	2.89 ^b^	4.96 ^e^	5.12 ^e^
**β-Myrcene**	26.66 ^a^	9.95 ^f^	10.34 ^e^	19.27 ^b^	10.78 ^e^	7.54 ^g^	13.03 ^d^	16.88 ^c^
**Limonene**	10.45 ^a^	2.17 ^e^	2.16 ^e^	4.13 ^b^	3.55 ^c^	1.58 ^g^	2.02 ^f^	2.77 ^d^
**β-*trans*-Ocimene**	1.12 ^a^	0.80 ^d^	0.62 ^e^	0.99 ^b^	0.50 ^f^	0.78 ^d^	0.68 ^e^	0.92 ^c^
**Terpinolene**	2.50 ^a^	0.97 ^e^	1.60 ^b^	1.66 ^b^	0.69 ^f^	0.29 ^g^	1.25 ^c^	1.29 ^c^
**β-(*E*)-Caryophyllene**	17.50 ^a^	25.35 ^e^	23.41 ^c^	17.91 ^a^	24.26 ^d^	22.62 ^b^	16.92 ^a^	24.24 ^d^
**α-Humulene**	7.26 ^a^	11.51 ^e^	9.62 ^c^	7.22 ^a^	9.23 ^c^	8.90 ^b^	8.41 ^b^	10.32 ^d^
**(*E*,*E*)-α-Farnese**	1.80 ^a^	1.82 ^a^	0.58 ^e^	1.05 ^d^	1.14 ^c^	1.44 ^b^	1.48 ^b^	1.03 ^d^
**Selina-4(15),7(11)-diene**	2.13 ^a^	1.21 ^f^	1.20 ^f^	1.35 ^e^	1.96 ^b^	1.43 ^d^	1.52 ^c^	1.47 ^d^
**Caryophyllene oxide**	3.79 ^a^	4.96 ^b^	7.82 ^c^	10.71 ^e^	11.04 ^e^	8.65 ^d^	12.16 ^f^	5.84 ^b^
**EO yield ^2^**	**0.21 ^a^**	**0.16 ^c^**	**0.14 ^d^**	**0.12 ^e^**	**0.18 ^b^**	**0.15 ^d^**	**0.15 ^d^**	**0.11 ^e^**
**% recovery of EO**	**100**	**76.19**	**66.67**	**57.14**	**85.71**	**71.42**	**71.42**	**52.38**

^1^ Dry mass calculated; ^2^ mL/100 g^−1^ according to distillation on Deryng apparatus; ^3^ values followed by the same letter within a row are not significantly different (*p* > 0.05, Duncan’s test).

**Table 5 foods-09-01118-t005:** Sensory profile of dried hemp flowers.

Aroma Description	Drying Method						
	CD 50 °C	CD 60 °C	CD 70 °C	VMD 240 W	VMD 360 W	VMD 480 W	CPD-VMFD
**Hemp ID**	3.0 ^c,†^	2.0 ^c,d^	1.0 ^d^	7.0 ^a^	2.0 ^c,d^	2.5 ^c,d^	3.0 ^c^
**Fresh vegetable**	4.5 ^c^	3.5 ^c^	1.0 ^d,e^	7.5 ^a^	2.0 ^d^	4.0 ^c^	3.5 ^c^
**Citrus**	3.5 ^b,c^	2.0 ^d^	1.0 ^e^	5.5 ^a^	2.0 ^d^	3.0 ^c^	2.5 ^c,d^
**Balsamic (rosemary)**	2.5 ^c^	2.0 ^c,d^	1.5 ^d,e^	5.5 ^a^	1.5 ^d,e^	2.5 ^c^	2.0 ^c,d^
**Spicy (black pepper)**	2.0 ^b,c^	1.5 ^c,d^	1.0 ^d^	3.5 ^a^	1.5 ^c,d^	1.5 ^c^	1.5 ^c,d^
**Anise**	2.5 ^b^	2.5 ^b^	1.5 ^c^	4.5 ^a^	1.5 ^c^	2.0 ^b,c^	1.0 ^c,d^
**Cooked**	0 ^c^	0 ^c^	0 ^c^	0 ^c^	0 ^c^	0 ^c^	0.5 ^b^
**Hay-woody**	1.0 ^c^	2.5 ^b^	2.5 ^b^	1.0 ^c^	3.0 ^b^	2.0 ^b,c^	2.0 ^b,c^
**Camomile**	1.0 ^c^	2.5 ^a,b^	2.5 ^a,b^	1.0 ^c^	2.5 ^a,b^	2.0 ^b^	2.0 ^b^
**Earthy**	0	0	0	0	0	0	0
**Burnt**	0 ^b^	0 ^b^	0 ^b^	0 ^b^	0 ^b^	0 ^b^	0 ^b^

^†^ Mean values followed by the same letter within the same row were not significantly different (*p* > 0.05) according to the HSD Tukey’s least significance difference test.

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
