# Peer review of "Volatile Composition and Sensory Properties as Quality Attributes of Fresh and Dried Hemp Flowers (*Cannabis sativa* L.)"

_foods, 2020, doi:10.3390/foods9081118_

Round 1

Reviewer 1 Report

-In Plant Material section particle size, density and distribution of material should be included.

-Based in other author studies, the equation that best fits the empirical model using drying kinetics is Midelli et al.: 

MR = b exp (-kt) + bT

A better clarification of why this equation has not been used would be necessary.

-Using the CD and VMD techniques, the use of the upper and lower limits of the study variables can be understood, as well as the use of a univariate analysis. However, with the CPD-VMFD technique, a multivariate study is necessary to be able to be compared with the rest in its best conditions.

Reviewer 2 Report

The range of drying conditions must be justified.

Why were temperatures from 50 to 70ºC selected?

Why were powers from 240 to 480 W selected?

Which is the justification to employ initial loading mass of sample to be dried? Was a monolayer of deep-layer configuration employed?

Why combined method was only tested in central conditions of temperature and power? Was this condition supported by the quality results found or attending to found results would be more adequate to test other conditions?

Really, an interesting result would be to report the optimal drying condition. Nevertheless, with current experimental planning this result is not accomplished.

Which is the evidence to neglect the equilibrium moisture content?

In the combined run: Which is the criterium to extend the convective drying until 60 min? Why not 40 or 80 min?

Line 129 must be Eq. (3).

Lines 132-134 must be rewritten.

Really the Page’s model is not employed throughout the manuscript. It must be consequently deleted.

Lines 159-161: please improve the specifications: a rate must be ºC/min (?). Last line is not comprehensible. From 150 to 270 at 15ºC/min, it does not correspond to 5 min.

Results:

Drying results (and modelling)

It is clear that drying modelling is not the main aim of the manuscript. Drying kinetics modelling is very simple and empirical. Nevertheless, authors must try to improve. Authors talk about “drying phases” for the analysis of the drying model parameters. The classical drying theory identifies a constant rate period (main mass transfer in gas or external phase) and falling rate period (solid or internal phase). The two terms model employed allow the estimation of external mass transfer coefficients (initial times), also diffusion coefficients at long times. Many references can be found about this in the bibliography. In example, results of Fig 1b for vacuum drying show clearly that the increase of power from 360 to 480 W is not appropriate because water removal rate is restricted by the mass transfer coefficient corresponding to external (gas) resistance. Please, re-discuss results attending these aspects.

One very important physical property during drying is the sample shrinkage (and intimately linked to drying rate) that in many cases is far from ideal shrinkage. Any evidence? Additionally, the drying rate is limited by the appearance of surface hardening. Any evidence for some drying conditions? Could both physical facts explain the results obtained with combined method?

The fit developed must contain some restrictions. In example, a +b = 1.

The number of figures of the parameters is not justified by the experimental errors during drying tests. Please reduce.

Parameters of the model have units. Please, add in Table and in the test.

Tmax shown in Table 1 is not analysed (or commented). This temperature could be interesting to explain some results. I understand that correspond the sample temperatures. Nevertheless, maximum temperature is not very relevant when it is not accompanied with when and the duration of the period that this temperature is achieved.

Regarding to colour results and discussion, authors must evaluate total colour differences among samples and conclude about significant or not differences.

Paragraph from lines 376 to 379 could be extended with deeply explanations and justifications about the results found. Relationships or some hypothesis with drying conditions could be established. It is the main aim of the paper!!

Line 389: Obviously is 60ºC.

Superscripts of Table 5 must be carefully revised.

Paragraphs from lines 398 to 410 must be deleted because is a repetition.

After all, how must I dry hemp flowers? Is it really established?

Reviewer 3 Report

The paper deals with the drying processes of hemp flowers (Cannabis sativa L.). In particular, the effects of the different drying methods on the content of volatile compounds and color are analyzed. Sensory analysis is also conducted to establish the influence of the processes on the perceived odor in the dried samples.

For the description of the moisture content profiles during the different drying processes, the empirical kinetic equation of Page was adopted.

The work, although interesting, has some important issues to be clarified:

1) Table 1 on page 7 does not show the units of measurement of parameters k1 and k2 (1/s? Or 1/min?).

There is no comment about these parameters, although 1/k1 and 1/k2 represent characteristic times of the "two phases" the authors divided the drying processes. The authors should comment adequately on these parameters (lines 250 259) and evaluate the duration of the two phases based on these rather than on a and b parameters.

Furthermore, in my opinion, the authors should clarify what the terms "first phase" and "second phase" exactly mean. Since Page's model is a purely kinetic, the phases the authors speak of (the first phase described by a*Exp(-k1*t), the second by b*Exp(-k2*t)) do not generally reflect a change in the physics of the problem (except perhaps in the case of the CPD-VMFD). Several works exist regarding mathematical modeling based on the physical principles of convective and microwave drying. The authors should add in my opinion comments about this.

Why was the CPD-VMFD process analyzed at T = 60 ° C and 360 W, only? Have these values ​​been selected based on an optimization study? If so, it is necessary to report the details of the optimization.

 It is also not clear at all why the CPD-VMFD process was introduced since there is not a comparison at different temperatures and powers values as for CD and VMD processes: from this point of view, the work seems to be incomplete.

Line 274. In my opinion, the comparison is wrong. The VMD 480W process has a maximum temperature of 61 ° C. It would, therefore, be more correct to compare it with the CD60 process or adequately justify the choice made. Why was it compared to CD50? Maybe the average temperature of the VMD 480W process is 50 ° C? If so, enter the average temperature values ​​in table 1 and enter the experimental procedure adopted on page 3 lines 109-116 and 117-120.

Eq.1, line 124 is not properly an equation in the mathematical sense but a relationship defining MR variable.

Line 132-134 and Eq. 4 The sentence is obscure for me. Moreover, Eq 4 is quoted and reported but it is not used throughout the text. Either you use it or you delete it.

Line 143 Enter the abbreviation EO for Essential Oils the first time the term is mentioned.

Line 304 Correct Table 3 instead of Table 2.

Table 4 on page 12 and lines 376-379. The values ​​shown in the table are not clear in my opinion. If I correctly understand, the table shows the percentage values ​​on a dry basis of the compounds but there are "anomalous" variations between fresh and dried. See, for example, the case of alpha-pinene: its percentage increases going from fresh to dried products. How is it possible? In general, I expect product degradation during drying processes. Do other chemical reactions take place? The percentage increase is due to something else? Please specify.

On table 4, the sum of percentages is less than 100%. Why? Are there other compounds, not mentioned? If so, in my opinion, it is necessary to specify them in the table (Insert for example a further raw “other compounds”). Please clarify this point in the table and the text.

Lines 398-410 The sentences are duplicated (see lines 381-393)

Line 390 The sentence is inaccurate since the CPD-VMFD process was performed only at a temperature of 60 ° C and a power of 360W

Line 103 and line 121 standardize the formatting of 10% wb (for example, that of line 95). What does wb mean? Please enter the meaning in parentheses the first use.

Conclusions are a bit poor. I would expect a critical comment on the results concerning the procedures adopted.

Round 2

Reviewer 1 Report

the explanation and implemented corrections are satisfactory to accept the manuscript in present form

Reviewer 2 Report

All my comments were conveniently replied.

Reviewer 3 Report

Just correct some typos